# Mechanisms Linking COPD to Type 1 and 2 Diabetes Mellitus: Is There a Relationship between Diabetes and COPD?

**DOI:** 10.3390/medicina58081030

**Published:** 2022-08-01

**Authors:** Sangmi S. Park, Jessica L. Perez Perez, Brais Perez Gandara, Christina W. Agudelo, Romy Rodriguez Ortega, Huma Ahmed, Itsaso Garcia-Arcos, Cormac McCarthy, Patrick Geraghty

**Affiliations:** 1Department of Medicine, State University of New York Downstate Health Sciences University, Brooklyn, NY 11203, USA; sangmi.park01@downstate.edu (S.S.P.); jessica.perezperez@downstate.edu (J.L.P.P.); brais.perezgandara@downstate.edu (B.P.G.); christina.agudelo@downstate.edu (C.W.A.); romy.rodriguezortega@downstate.edu (R.R.O.); huma.ahmed@downstate.edu (H.A.); itsaso.garcia-arcos@downstate.edu (I.G.-A.); 2University College Dublin School of Medicine, Education and Research Centre, St. Vincent’s University Hospital, D04 T6F4 Dublin, Ireland; cormac.mccarthy@ucd.ie

**Keywords:** chronic obstructive pulmonary disease, diabetes, hyperglycemia, inflammation, oxidative stress, insulin, metabolism

## Abstract

Chronic obstructive pulmonary disease (COPD) patients frequently suffer from multiple comorbidities, resulting in poor outcomes for these patients. Diabetes is observed at a higher frequency in COPD patients than in the general population. Both type 1 and 2 diabetes mellitus are associated with pulmonary complications, and similar therapeutic strategies are proposed to treat these conditions. Epidemiological studies and disease models have increased our knowledge of these clinical associations. Several recent genome-wide association studies have identified positive genetic correlations between lung function and obesity, possibly due to alterations in genes linked to cell proliferation; embryo, skeletal, and tissue development; and regulation of gene expression. These studies suggest that genetic predisposition, in addition to weight gain, can influence lung function. Cigarette smoke exposure can also influence the differential methylation of CpG sites in genes linked to diabetes and COPD, and smoke-related single nucleotide polymorphisms are associated with resting heart rate and coronary artery disease. Despite the vast literature on clinical disease association, little direct mechanistic evidence is currently available demonstrating that either disease influences the progression of the other, but common pharmacological approaches could slow the progression of these diseases. Here, we review the clinical and scientific literature to discuss whether mechanisms beyond preexisting conditions, lifestyle, and weight gain contribute to the development of COPD associated with diabetes. Specifically, we outline environmental and genetic confounders linked with these diseases.

## 1. Introduction

Lifestyle risk factors are considered central to the development of type 1 and 2 diabetes mellitus (T1D and T2D) and chronic obstructive pulmonary disease (COPD) [1]. The daily physical activity of COPD patients is reduced in the early phases of the disease, as compared with healthy age-matched controls, and worsens over time [2]. Poor medication adherence is described in patients with these diseases, resulting in increased hospitalization rates. However, a series of clinical studies described herein link COPD and T1D and T2D. Equally, studies in disease models provide mechanistic evidence to suggest that comorbid diabetes and COPD feedback influence the progression of the other disease. This review will focus on the epidemiology, physiology, molecular data, and disease models linking diabetes and COPD.

## 2. Evidence linking COPD to Diabetes

### 2.1. Epidemiological Evidence Linking COPD to T1D

The prevalence of T1D was previously reported to be increasing over the past decade [3]. Reduced total lung capacity (TLC), diffusing capacity of the lung for carbon monoxide (DLCO), pulmonary elastic recoil, and end-expiratory lung volume are detected in patients with T1D. This impaired lung physiology is inversely correlated with glycated hemoglobin levels [4]. Changes in collagen glycation of lung parenchyma and alveolar microangiopathy may contribute to this altered pulmonary dysfunction. In a small comparative study, T1D patients exhibited normal spirometry and pleural pressure, but a higher dynamic elastance during hypoxia, possibly indicating peripheral airway involvement [5].

### 2.2. Epidemiological Evidence Linking COPD to T2D

T2D is a leading comorbidity in COPD [6,7]. A population-based retrospective study from Italy demonstrated a higher prevalence of T2D in COPD patients (18.7%) compared to the general population (10.5%) [6]. In this study, women with COPD were significantly more likely to develop T2D compared to women without COPD [6]. Another population-based study in Taiwan [7] showed that T2D was present in 16% of patients with COPD, and within a 10-year follow-up period, T2D was newly diagnosed in 19% of COPD patients, showing increased prevalence and incidence of the disease. Additionally, the association between diabetes and pulmonary disease did not extend to asthma, according to one prospective cohort study [8], suggesting a specific interplay between COPD and diabetes.

Hyperglycemia is an independent predictor of poor outcomes in patients admitted to the hospital and intensive care unit (ICU) [9]. In a study looking at patients admitted with acute decompensated respiratory failure complicating COPD, baseline hyperglycemia upon presentation was identified as a good predictor of clinical outcomes, determined by the Acute Physiology and Chronic Health Evaluation II (APACHE II) score [10]. Mortality rates are high in patients with COPD, as demonstrated by death in almost 80% of patients within nine years of hospital admission due to acute exacerbation of COPD, and diabetes was associated with decreased long-term survival in these patients [11]. Diabetes and cardiovascular diseases were associated with increased mortality in a cohort of COPD patients, when adjusted for age, gender, and smoking pack-year history [12].

### 2.3. T1D Affects Specific Lung Function Parameters

T1D is associated with decreased TLC, lung elastic recoil, diffusion capacity to transport carbon monoxide (DLCO), and pulmonary capillary volume [13,14]. These changes in pulmonary function were present, even in the absence of established pulmonary disease. Non-smokers with T1D who were not previously diagnosed with the pulmonary disease had decreased distance in the 6-minute walk test, forced expiratory volume in one second (FEV1), TLC, and DLCO [15]. Poor glycemic control, duration, and severity of diabetes were associated with worsening lung function, observed by changes in forced vital capacity (FVC) and FEV1 [16,17,18]. Patients with higher hemoglobin A1c (HbA1c) have lower FVC, FEV1, vital capacity, and peak expiratory flow (PEF) [19]. These abnormalities can be mitigated in just three months after correction of hyperglycemia [20].

Systemic inflammation plays a significant role in the pathogenesis and progression of COPD and diabetes. C-Reactive Protein (CRP) levels are inversely associated with FEV1 and FVC at baseline [21]. These changes are present in both sexes and are independent of smoking, obesity, and the presence of other respiratory pathologies, such as asthma [22]. Lung responses appear to be altered by complications of diabetes, with impaired autonomic nerve function in the lungs of T1D patients [23]. In a study testing the responses of diabetic subjects and non-diabetic controls to hypoxia, hypercapnia, and exercise, approximately 25% of diabetic subjects had evidence of impaired sensitivity to hypoxia or decreased ventilatory response to hypercapnia [24]. More recently, the approach to evaluating autonomic dysfunction using assessing cardiorespiratory function has created a body of evidence that proposes that these abnormalities could potentially be corrected with new interventions [25].

### 2.4. T2D Affects Specific Lung Function Parameters

The alveolar microvascular function is impaired in T2D non-smokers compared to controlled subjects, as demonstrated by decreased DLCO [26]. When using the German COPD and Systemic Consequences–Comorbidities Network (COSYCONET) cohort, hyperlipidemia (prevalence of 42.9%) is associated with lower intrathoracic gas volume and higher FEV1, when adjusting for risk factors and other comorbidities [27].

### 2.5. Metabolic Syndrome in COPD

Metabolic syndrome (MetS) represents a major public health challenge and confers a five-fold increase in the risk of T2D and a two-fold increase in the risk of developing cardiovascular disease (CVD) within five to ten years [28]. MetS is defined by a constellation of closely related cardiovascular risk factors, including obesity, altered lipids, increased blood pressure, and impaired fasting glucose [28]. A recent cohort of 7358 adults described the association between MetS and pulmonary function. The risk of MetS was higher in patients with airway obstruction than in those without (odds ratio (OR) 1.47; confidence interval (CI) 1.12–1.92), and after adjusting for body mass index (BMI), central obesity was significantly associated with airflow obstruction (OR 1.43; 95% CI 1.09–1.88) [29].

According to the International Diabetes Confederation, neither COPD nor cigarette smoking was included as fundamental risk factors of MetS. However, an increased prevalence of MetS is observed in COPD patients compared to the general population (21–62%) [29,30,31,32,33]. In particular, patients with earlier stages of COPD exhibit the highest prevalence of MetS [30,31]. In the general population, MetS becomes more prevalent with increasing age [34]. COPD patients with MetS often display worsened courses of the disease, as observed by greater percent-predicted FEV1 reduction, increased dyspnea, and greater use of inhaled steroids [35]. Two COPDGene studies found that diabetes is more frequent in subjects with airway disease than emphysema on CT [36,37]. Therefore, it is warranted to monitor COPD patients without emphysema for diabetes, hypertension, and hyperlipidemia.

### 2.6. Cigarette Smoking in Diabetics

In a cohort study detailed by George et al., after calculating the attributable risk of COPD, cigarette smoke (CS) accounted for 19% of cases in T1D and 30% of cases in T2D, compared to 26% of cases in non-diabetics [38]. While it is important to note that MetS and hyperglycemia are also described as risk factors for reduced lung function in healthy non-smoking subjects [39], CS may nonetheless play a role in the pathophysiology of COPD in diabetics. A recent large-scale cross-trait GWAS paper investigating genetic overlap between COPD and several cardiac traits (resting heart rate, high blood pressure, coronary artery disease, and stroke) from the UK Biobank, the CARDIoGRAMplusC4D Consortium, and the International Stroke Genetics Consortium demonstrated smoke-related single nucleotide polymorphisms (SNPs) located in the 15q25.1 region that were associated with cigarette smoke usage, resting heart rate, and coronary artery disease [40]. This region was also linked to COPD in a separate study [41]. It is suggested that this smoke-related 15q25.1 region may play a role in the severity of nicotine, alcohol, and opioid dependence [42], partially due to it containing three nicotinic cholinergic receptor genes (*CHRNA5*-*B4*). A non-synonymous single-nucleotide polymorphism of *CHRNA5*, rs16969968, can result in impaired ciliogenesis and the altered production of inflammatory mediators in airway epithelial cells [43]. Cigarette smoke exposure can also influence differential methylation of CpG sites on genes linked to T2D, such as *ANPEP*, *KCNQ1*, and *ZMIZ1* [44].

### 2.7. Alpha-1 Antitrypsin and Diabetes

Several clinical trials were undertaken to investigate the potential for alpha-1 antitrypsin (AAT) infusions as a treatment for diabetes [45,46,47], specifically T1D. Raising blood levels of AAT with augmentation therapy is reported to prevent T1D development, prolong islet allograft survival [48], increase insulin release capacity [49], and inhibit pancreatic β-cell apoptosis [50]. AAT treatment also significantly reduces HbA1c levels [51]. There is an association between AAT deficiency with an increased risk of developing T2D [52]. High levels of degraded AAT are observed in the urine of T2D patients with diabetic kidney disease [53]. We recently published a review focusing on AAT and diabetes, which can be accessed for further reading on this topic [54].

## 3. Animal Models of T1D and T2D in COPD

There are many studies investigating obesity and pulmonary diseases, but here, we will only discuss T1D and T2D models with noted pulmonary involvement. Alloxan-induced T1D rats are more susceptible to emphysematous lesions in response to porcine pancreatic elastase (PPE) instillation compared to nondiabetic control mice [55]. The diabetic rats had a reduced number of neutrophils in the bronchoalveolar lavage fluid (BALF) and diminished repair of the alveolar walls in response to emphysema. Insulin treatment restored these changes in neutrophil numbers and the magnitude of emphysematous lesions [55]. Mice with cystic fibrosis-related diabetes (CFRD) have increased blood glucose concentration, which is associated with impairment in bacterial clearance from the lung in diabetic mice [56]. Streptozotocin (STZ)-induced hyperglycemia in rats results in lung oxidative stress, as well as changes in lung structure and gas exchange [57]. The same pathomorphological modifications of the lungs, including thickening of the alveolar–capillary barrier, collapsed alveolar epithelium, and destruction of the matrix, are observed in STZ-induced hyperglycemia in hamsters [58]. The structural modifications were more pronounced and developed at a faster rate in hamster models of diabetes associated with hyperlipidemia. Other research groups demonstrated that STZ-induced T1D in rats results in a pulmonary fibrosis phenotype [59,60]. Therefore, hyperglycemia associated with diabetes likely contributes to the pathophysiology of lung diseases.

## 4. Mechanistic Link between COPD and T1D

Diabetic patients exhibit thickening of the pulmonary basal lamina [61]. Furthermore, T1D patients who have never smoked exhibited thickening of the alveolar-capillary membrane [62]. These morphological features are commonly found in diabetic microangiopathy and hence, may explain the decrease in DLCO in T1D. Similar histopathological pulmonary changes are also observed in experimental models of T1D [63]. Aside from morphological changes, fibrotic changes in the small airways due to chronic inflammation can progressively result in COPD. TGF-β can trigger fibrotic changes in the lungs in human patients and experimental T1D models [59,60]. While T1Ds are more susceptible to impaired lung function and structural changes leading to COPD [17,38,64], the exact mechanisms underlying the association between the two diseases are not yet known. Here, we will outline possible mechanistic links between these diseases (see Figure 1).

### 4.1. Oxidative Stress

Oxidative stress is a common feature of COPD and T1D pathogenesis and is caused by an imbalance in oxidant-antioxidant levels. Chronic exposure to CS and other particulate pollutants induces oxidative stress in the lungs [65]. Alveolar macrophages from individuals with COPD generate increased superoxide radicals and hydrogen peroxide compared to those without COPD [66]. Lipid peroxidation (a marker of oxidative stress) and its by-products are elevated in both lung tissue and exhaled breath condensate in individuals with COPD [67]. In addition, BALF concentrations of the antioxidant glutathione are lower in exacerbators compared with individuals with stable COPD [68]. Increased reactive oxygen species (ROS) and oxidative stress can result in impaired efferocytosis, or immune cell-mediated clearance of apoptotic cells, as CS exposure decreases the clearance of apoptotic cells by alveolar macrophages through activation of Rho GTPases [69].

Oxidative stress is reported primarily in the early stages of T1D and is a suggested mechanism that contributes to glucose autoxidation, glycation of proteins, consumption of NADPH through the polyol pathway, and activation of protein kinase C [70]. Superoxide dismutase (SOD) activity and increase in nitric oxide (NO) are decreased in the lungs of animal models of T1D. Oxidative stress appears to remain, even when hyperglycemia is controlled [71], suggesting that oxidative stress may be an important factor early on in the development of T1D.

### 4.2. Metabolic Changes in T1D and COPD

T1D involves multiple metabolic alterations and their associations with pulmonary disease, and specifically with COPD, have been the subject of intense study. However, the complexity of in vivo metabolism poses an intrinsic difficulty for the interpretation of the available data. Serum concentrations of triglycerides and major choline-containing phospholipids, such as sphingomyelins, phosphatidylcholines, and phosphatidylethanolamines, are reduced in infants who later develop T1D [72], likely reflecting defects in lipoprotein and glucose metabolism in hepatic and peripheral tissues. Plasma sphingomyelin is negatively correlated with COPD, while glycosphingolipids are associated with COPD exacerbations [73].

Thus far, 168 sphingolipids and 36 phosphatidylethanolamine lipid levels are reportedly elevated in the sputum of smokers with COPD compared to smokers without COPD [74]. Among these, the expression of 13 lipids involved in the sphingolipid pathway (4 ceramides, 5 dihydroceramide, 1 sphingomyelin, and 3 glycosphingolipids) was increased by 2.5-fold or more and showed an inverse correlation with lung function. Because the lipids collected in sputum originate mostly in the upper airways, these data may reflect the increased production of ceramides associated with inflammation. Indeed, ceramides are involved in the destruction of the alveolar integrity in emphysema [75]. Amongs the lipids detectable in the distal areas of the lung, the alveolar availability of surfactant PC and of sphingomyelin correlates with FEV1 in COPD patients [76].

Furthermore, lysoPC is decreased in rats subjected to CS and LPS instillation, and this alteration was associated with the onset and development of COPD [77]. LysoPC also amplifies PPE-induced emphysema in hamsters [78]. LysoPC may have the potential to activate inflammatory responses [79], which might explain the changes in lysoPC levels observed in inflammatory diseases, including T1D and COPD. For further reading on lipids in pulmonary diseases, see this recent review from our group [80].

### 4.3. Immune Responses in COPD and T1D

Several immune responses are implicated in the pathogenesis of COPD and T1D, particularly Toll-like receptors (TLR) [81,82,83]. The respiratory tract is vulnerable to infections and exposure to environmental factors, such as CS, modulates immune responses to infections [84]. TLR2, TLR3, and TLR4 are frequently implicated in the pathogenesis of diabetes, T1D-related vascular complications, and CS-induced inflammation and COPD [83,85].

Circulating levels of endotoxin and monocytes expressing TLR2 and TLR4 are elevated in T1D patients [86,87], possibly due to high glucose and low insulin levels in T1D patients [88]. Likewise, TLR2 and TLR4 expressions are increased in alveolar macrophages of COPD patients [89] and the airways of CS-exposed mice [90]. Biobreeding diabetes-prone rats, a genetic model of T1D, have increased diabetogenicity upon infection with the Kilham rat virus in combination with a TLR3 ligand [91]. In T1D, nuclear factor kappa B (NF-κB) activation in the β-cells is triggered by TLR-mediated pathways [92]. Equally, *Tlr2* knockout mice are resistant to STZ-induced autoimmune diabetes. Non-obese diabetic (NOD) mice (a T1D model), deficient in TLR2 or treated with agonistic TLR4 monoclonal antibody, are resistant to new-onset T1D due to antigen-presenting cell tolerance and compromised adaptive T cell response [93]. CS exposed animals exhibited airway fibrosis, alveolar enlargement, and decreased lung function, and these changes are attenuated in *Tlr4*^-/-^ mice [90]. TLR2 expression is increased in monocytes in smokers [94].

Hypoxia enhances systemic inflammation through diffuse activation of NF-κB [95]. Plasma concentrations of inflammatory markers, including serum CRP, IL-6, IL-1β, and TNF-α, are elevated in T1D patients, which contribute to pathological processes of diabetic complications, such as microvascular complications [96]. Similarly, COPD exacerbates systemic inflammation in smokers, as indicated by elevated circulating concentrations of leukocytes, CRP, fibrinogen, and TNF-α, and these inflammatory processes continued, even after smoking cessation [97]. Importantly, CRP is associated with a decline in lung function in young adults [22]. This alteration in the inflammation factor may cause tissue damage, which may explain lower pulmonary DLCO observed in children with T1D [98]. DLCO impairment may be an early marker of T1D-related complications, such as microangiopathy [98,99]. T1D and its complications, particularly microangiopathy, are associated with pulmonary function changes. Similarly, the presence of systemic inflammation in COPD is linked with a variety of comorbidities, such as cardiovascular diseases [100]. The increased serum levels of IL-6 and plasma fibrinogen concentrations observed in COPD exacerbation predispose individuals to cardiovascular disease development [101]. Finally, the 15q25.1 region contains the cathepsin H gene (*CTSH*), and *CTSH* expression is suppressed by cytokines in islets and β-cells [102]. However, if CTSH is overexpressed in insulin-producing β-cells (INS-1 cell line), it protects the β-cells against cytokine-induced apoptosis [102]. CTSH is also a positive regulator of insulin transcription, and the *CTSH* SNP, rs3825932, affects the expression level of CTSH linked to β-cell function in T1D.

### 4.4. Impaired Wound Healing

Exposure to CS can alter the repair mechanisms within the lungs, leading to small airway remodeling [103]. Likewise, wound healing is compromised in T1D. Repetitive injury of the airway epithelium by CS also upregulates TGF-β in COPD patients, which subsequently leads to persistent activation of epithelial to mesenchymal transition (EMT) and airway remodeling [104]. Airway epithelial cell senescence may also contribute to the pathogenesis of COPD by inhibiting wound repair and increasing the release of pro-inflammatory cytokines involved in p38-MAPK activation [105]. Fibroblasts from COPD patients are less capable of sustaining tissue repair, which may be due to decreased synthesis and deposition of extracellular matrix (ECM) components, as well as decreased sensitivity to TGF-β1 [106]. Additionally, the loss of regenerative capability of the bronchial progenitor cells in CS-induced COPD could contribute to airway remodeling and impaired wound repair [107,108].

Studies in T1D patients and animal models demonstrated decreased collagen deposition and delayed wound repair responses [109]. This decreased reparative collagen accumulation in response to wounds coincides with decreased nitric oxide (NO) synthesis in wounds [110], decreased fibroblast proliferation [109,111], and the elevated plasma level of glucocorticoids [112]. Decreased tensile strength [113], reduced expression of IGF-1 and TGF-β in wound fluid [114], and dysregulated late inflammatory response [115] are also suggested to contribute to the impairment of wound healing in T1D in parallel with decreased collagen deposition. Decreased proliferative potential and migration of adipose tissue-derived stem cells (ASCs) may also lead to inefficient wound healing in T1D [116].

### 4.5. Glucose and COPD

Hyperglycemia can also contribute to airway inflammation [64]. Experimental rat models of T1D have decreased glucose oxidation and a decreased rate of glucose incorporation into neutral lipids and phospholipids [117,118]. Glucose metabolism also plays an important role in surfactant metabolism. The dysregulation of glucose observed in the T1D rat model may potentially lead to the dysregulation of pulmonary surfactant synthesis [119], as there are reduced lamellar inclusion bodies in type II alveolar epithelial cells in the T1D rat model, which are a source of pulmonary surfactant [119]. While studies have looked at the transport of glucose in lung cells [120], in vitro models studying the effect of extracellular glucose on airway cell responses may serve as useful tools for delineating the association between hyperglycemia and COPD. Finally, pancreas and kidney transplantation ameliorated the decline in lung function, in particular FEV1 and FVC, in a T1D population [121]. This suggests that the reversal of T1D influences pulmonary outcomes. In a longitudinal study where 7055 participants were followed over 3.9 years, a longitudinal change in HbA1c was associated with a longitudinal change in FVC. Participants with new-onset diabetes had a higher decline in FVC over time [122]. Equally, diet quality is associated with better FEV1, FVC, and restriction [123].

## 5. Mechanistic Links between COPD and T2D

### 5.1. Oxidative Stress

T2D subjects show increased ROS generation and markers of oxidative stress, as well as decreased antioxidant levels compared to non-diabetics [124]. At the cellular level, oxidative stress negatively regulates insulin signaling via interactions with serine/threonine kinases [124], contributing to insulin insensitivity. Systemically, oxidative stress decreases pancreatic β-cell insulin secretion and subsequently impairs insulin signaling in peripheral tissues [125]. Furthermore, pancreatic β-cells are uniquely susceptible to oxidative stress [125]. CS exposure decreases β-cell insulin production, processing, and secretion, and decreases β-cell enrichment and proliferation. CS increases β-cell oxidative and endoplasmic reticulum stress and induces ceramide accumulation [126].

### 5.2. Inflammation

Similar to T1D, chronic inflammation is observed in T2D. Individuals with obstructive lung diseases have significantly elevated abdominal adipose tissue [127], and obesity is common in the early stages of COPD [128]. Adipose tissue inflammation is also present in individuals with mild-to-moderate COPD [129]. Furthermore, individuals with COPD have higher levels of plasma CRP compared to control subjects, and CRP levels positively correlate with macrophage infiltration of adipose tissue upon biopsy [129]. Various adipokines, including adiponectin, are associated with worse outcomes in COPD [130]. However, this study had significant confounding factors, with differences in sex, age, pack-years, BMI, methacholine responsiveness, and ethnicity among its group. Obesity increases levels of adipokines and alters cellular immunity. A recent cross-trait genome-wide association study (GWAS), using 457,822 subjects of European ancestry from the UK Biobank, found a positive genetic correlation between BMI and later-onset asthma, with onset at or after 16 years of age [131]. They identified 34 shared loci among 3 obesity-related traits and 2 asthma subtypes and, utilizing an obesity mouse model, identified 2 genes (acyl-coenzyme A oxidase-like (ACOXL) and myosin light chain 6 (MYL6)) playing a significant role in both diseases. Equally, a recent GWAS study utilizing data on 100,285 subjects from the China Kadoorie Biobank (and the UK Biobank) identified 9 novel loci for FEV1, 6 for FVC, and 3 for FEV1/FVC linking lung function to obesity [132]. The biological pathways linking lung function to obesity were cell proliferation; embryo, skeletal, and tissue development; and regulation of gene expression. This study also suggested that BMI had a negative effect on lung function over an eight-year follow-up [132]. These studies suggest that genetic predisposition, in addition to weight gain, can influence lung function. In addition, individuals with T2D are more susceptible to infection, possibly due to the increased risk of skin barrier disruption from diabetic neuropathy and diminished cellular immunity, including suppressed cytokine production and defective phagocytosis [133].

Insulin resistance could contribute to this diminished cellular immunity in T2D, as insulin administration decreases infection and complication rates in T2D populations [134]. Insulin has anti-inflammatory effects and suppresses ROS production [135]. Insulin also acts through PI3 kinase and ERK pathways to increase the secretion of IL-6 and TNF-α from activated LPS-treated macrophages [136]. Insulin therapy also has direct effects on the lung, inducing prostaglandin-mediated airway smooth muscle contraction [135]. Furthermore, the incretin glucagon-like peptide 1 (GLP-1), responsible for raising insulin during meals, may play a role in ROS and influence the responses for the receptor for advanced glycation end-products (RAGE). GLP-1 receptor agonists have anti-inflammatory benefits in the treatment of bronchial hyperresponsiveness [137] and obesity-related asthma [138]. Additionally, the nod-like receptor, containing a pyrin domain 3 (NLRP3) inflammasome, is linked to the development of COPD and other pulmonary conditions [139]. NLRP3 is significantly upregulated in an in vitro model of COPD exacerbation [140]. NLRP3 also participates in obesity-induced inflammation and induces insulin resistance [141].

Inflammation in the lung affects peripheral energy utilization. For instance, NF-kB-activation in the lung attenuates the suppression of hepatic glucose production by insulin and induces insulin resistance in peripheral tissues [142]. Inflammation in the lung may negatively affect systemic glucose homeostasis by decreasing the recruitment of skeletal muscle capillaries that deliver glucose and insulin to myocytes, thereby increasing blood glucose levels [143].

### 5.3. Immunometabolism in COPD and T2D

While the exact mechanism by which CS induces insulin sensitivity is not known, metabolic changes that may lead to insulin sensitivity are observed in smokers. Serum glucose, insulin, and C-peptide levels increase in response to glucose tests in cigarette smokers compared to nonsmokers [144]. Moreover, smokers with T2D have increased plasma insulin and triglyceride levels and decreased HDL cholesterol, [144]. Hypertriglyceridemia is associated with insulin resistance [145] and could be one of the contributing factors to developing insulin resistance in smokers. CS is suggested to impair insulin sensitivity in healthy and T2D subjects [144].

There are conflicting results on the effects of CS on adipokine regulation. While some studies demonstrated a direct association between alveolar and plasma adiponectin with CS [146,147], other studies showed that serum adiponectin levels are negatively associated with CS [148], or are not affected by CS status [130]. These differences in adiponectin associations may be due to multiple differences between study groups, such as sex, age, pack-years, BMI, methacholine responsiveness, and ethnicity. Likewise, some studies reported reduced leptin levels in smokers compared to nonsmokers [149,150], whereas another study suggested that nicotine may have a direct effect on insulin resistance by increasing circulating leptin levels [151]. Whether CS directly affects adipokine secretion in COPD patients remains to be elucidated.

There may be an association between metabolism and immunity when linking obesity and low-grade inflammation [152]. Immune cells, such as mononuclear cells of COPD patients and the mouse macrophage RAW264.7 cell line exposed to CS condensate (CSC), show impaired glycolysis and fatty acid oxidation [153]. Furthermore, mitochondrial ROS production is increased in CSC-exposed RAW264.7 cells in a dose-dependent manner. Immunometabolic changes are likely due to mitochondrial damage induced by CS exposure, which may participate in the progression of COPD [153]. Dysfunctional mitochondria with increased production of ROS are also observed in T2D [154]. Alveolar macrophages in COPD patients exhibit mitochondrial dysfunction [155], and macrophages with such metabolic impairment show reduced ability to migrate, phagocytose, and clear bacterial load [156], emphasizing the importance of metabolic regulation by immune cells to maintain lung homeostasis.

The role of adipose tissue in inducing systemic inflammation in obese COPD patients may contribute to insulin resistance [129,157]. Adipose tissue is proposed to actively participate in enhancing low-grade systemic inflammation in the lungs by producing adipokines, such as leptin and adiponectin, that regulate energy metabolism and inflammatory signaling. Leptin and adiponectin levels in adipose tissue positively correlated with circulating levels of the two adipokines and inflammatory markers in obese individuals, suggesting that adipocytokine secretion in obesity can contribute to systemic inflammation [158]. The circulating levels of adiponectin are increased in COPD patients compared to healthy subjects and are inversely associated with lung function, in particular FEV1 and FEV1/FVC, in COPD patients [146,147].

Increased adipose tissue mass is associated with susceptibility to pulmonary infection [157]. Adipose-derived proinflammatory cytokines, such as TNF-α, could inhibit insulin receptor signaling [159]. Circulating leptin levels positively correlate with adipose mass [129]. Leptin, is present in the induced sputum of COPD patients and is proposed to be involved in the regulation of the innate immune system of the lungs [160]. Leptin is also elevated in the sub-mucosa of COPD patients and negatively correlates with FEV1 and FEV1/FVC [161]. Plasma leptin, CRP, and Hb1Ac levels are elevated in COPD patients and are associated with cardiometabolic complications [162]. COPD patients exhibit increased ectopic fat accumulation, which is reported to cause insulin resistance and T2D [162].

### 5.4. Hyperglycemia and Hyperinsulinemia

High sucrose intake negatively affects lung mechanics and alveolar septal composition in mice, causing pulmonary extracellular matrix remodeling and reduced elasticity [163]. Hyperglycemic individuals exhibit elevated glucose concentrations in nasal secretions compared to normoglycemic individuals [164]. High airway glucose may foster a favorable environment for microbial colonization, with glucose in bronchial aspirates increasing the risk of respiratory methicillin-resistant Staphylococcus aureus in intubated patients [165]. Hyperglycemia can enhance the contractility of airway smooth muscle via the Rho-associated-coil-containing protein kinase pathway, intracellular calcium releases, and phosphorylation of myosin-targeting subunit-1 [166].

AGEs are formed as a consequence of chronic hyperglycemia via non-specific glycation and deposited throughout the body, leading to serious complications, such as endothelial cell disruption with subsequent micro-and microvasculature damage and, ultimately, organ failure [167]. In the lung, microvascular and parenchymal changes induce systemic hypoxia and dysregulate energy metabolism. AGEs also exert potent signaling activity when bound to RAGEs. AGE-RAGE complexing induces the expression of inflammatory genes in target cells via the NF-kB pathway [168]. Similar to T1D, sustained activation of the NF-kB pathway is observed in T2D [169]. In addition, AGEs also increase plasma CRP and TNFα secretion from mononuclear cells in individuals with T2D [170]. RAGE is overexpressed in the airway epithelium and smooth muscle of patients with COPD [171]. The role of AGE and RAGE in T2D and COPD require further exploration, but represents mechanistic areas of interaction for both diseases. Recently, a meta-analysis suggested that lower soluble RAGE is a biomarker for the presence of emphysema and airflow obstruction [172].

### 5.5. Autonomic Dysregulation

Autonomic dysregulation may be attributed to genetic factors independent of glucose and insulin levels, impacting both COPD and T2D pathogenesis. The B2-adrenergic receptor gene (ADRB2) encodes a multifunctional protein important for smooth muscle relaxation in the lung [173] and insulin secretion from the pancreas [174]. Specific polymorphisms in ADRB2 may lead to the co-occurrence of COPD and T2D. The R16G ADRB2 polymorphism correlated with decreased FEV1 and COPD severity [175], and also correlated with insulin sensitivity in an obese postmenopausal cohort of women in another study [176]. Other polymorphisms of the ADRB2, including T164I, are also known to have systemic consequences, including poor cardiovascular health and outcomes [177], which could affect COPD and T2D pathogenesis. Thus, ADRB2 may present a unique candidate for investigation of the mechanistic link between COPD and T2D co-occurrence in a subset of individuals (see Figure 2).

## 6. Therapy

There are several potential therapeutic strategies common for COPD and diabetes treatment. Here, we briefly discuss these treatment options, such as metformin, corticosteroids, thiazolidines, and AAT augmentation therapy (see Table 1 for a summary).

### 6.1. Metformin

Metformin, a biguanide antidiabetic drug, is recommended as the first-line therapy for T2D [202] due to its efficacy, relative safety, and beneficial effects of reducing HbA1C levels and weight, in addition to its general tolerability and favorable cost [202]. Moreover, it reduces cardiovascular (CV) mortality, all-cause mortality, and CV events in T2D patients with coronary artery disease (CAD), but not in non-diabetic patients with CAD or with a history of myocardial infarction (MI) [180]. Interestingly, a recent retrospective study demonstrated that metformin treatment for 2 years improved survival rates in COPD patients with T2D [184]. Equally, Mendy et al. found a reduction in mortality of patients with chronic lower respiratory diseases treated with metformin [203]. Metformin was piloted as therapy for many conditions outside of diabetes, including treatment of severe COPD exacerbations [187]. Metformin inhibits proinflammatory NF-kB signaling in human vascular wall cells [204], potentially dampening lung microvascular complications of T2D. Metformin improves glycemic control in T2D patients and therefore, reduces the formation of AGEs, but additionally, it is an effective scavenger of AGEs [205]. Another recent animal study suggested that activation of 5′-adenosine monophosphate-activated protein kinase (AMPK) by metformin could reduce abnormal inflammatory responses in mice with elastase-induced emphysema, as well as cellular senescence [179].

In COPD, changes to the aero-digestive microbiome are apparent and are associated with disease progression and exacerbations [206]. Metformin is known to change the composition of gut microbiota, induce improved insulin resistance, and decreased tissue inflammation [181]. Additionally, metformin reduces the frequency of lung infections, as demonstrated by the reduction in the glucose-induced growth of *Staphylococcus aureus* [207]. A study by Wishwanath et al. [182] highlighted the potential use of metformin to reduce the hyperglycemia-induced growth of *Pseudomonas aeruginosa*. Metformin was also found to enhance the macrophage bactericidal activity and improve survival in *Legionella pneumonia* [183]. Osteoporosis is more prevalent in patients with advanced COPD due to the direct effect of inhaled or oral corticosteroids. Metformin has anti-inflammatory properties and can decrease the prevalence of osteoporosis in patients with GOLD group D COPD [208].

Metformin may improve health status, symptoms, hospitalizations, and mortality in patients with COPD and T2D [188]. In an unmatched cohort study in Taiwan, T2D patients who had used metformin as an antidiabetic agent were less likely to develop COPD, with a hazard ratio (HR) of 0.56 (95% CI 0.537–0.584) [209]. In a prospective open-label trial of patients with moderate and severe COPD who also had T2D, the use of metformin showed improvement in symptoms and transitional dyspnea index scores compared to the baseline [186]. However, physiological outcomes, including PFTs and exhaled nitric oxide, were unchanged in this study [186]. Metformin was also studied regarding COPD exacerbations in patients without T2D, but failed to demonstrate any improvement in blood glucose control, nor effects on CRP or clinical outcomes in the non-diabetic population [187]. A recent study in Taiwan suggested that metformin use in patients with T2D and COPD was associated with higher risks of pneumonia, hospitalization for COPD, and invasive mechanical ventilation [189]. However, a recent observational study demonstrated that metformin use was associated with lesser emphysema progression over time in humans [185,210], possibly due to metformin protecting against smoke-induced lung, renal, and muscle injury, mitochondrial dysfunction, and ER stress in mice [210].

### 6.2. Additional Therapeutic Approaches

Inhaled corticosteroids are among the first line of treatments for COPD exacerbation, which leads to rapid loss of lung function and hospitalization. In patients with diabetes, inhaled corticosteroid use is associated with a dose-dependent elevation in serum glucose concentration [191] and an increase in HbA1c [192]. While systemic corticosteroid use is a known risk factor for the development of T2D, the risks harbored by inhaled corticosteroid use, which involves more localized administration, remain debatable. An increased risk of new-onset diabetes was reported in patients taking inhaled corticosteroids [193,194]. The current use of inhaled corticosteroids in patients with pulmonary disease was associated with a 34% increase in the rate of diabetes and an increased rate of diabetes progression [195]. This risk was greatest with the highest inhaled corticosteroid doses, equivalent to fluticasone 1000 mg per day, or more. However, another study investigating the effect of inhaled corticosteroid treatment did not find an increased risk of diabetes among current users [196]. Another cross-sectional, single-center study was performed in adults with established COPD to determine the prevalence of dysglycemia in COPD patients. The study utilized the oral glucose tolerance test to detect dysglycemia in COPD patients and revealed a near doubling of dysglycemia. The incidence of newly diagnosed diabetes was 21.7% and prediabetes was 30.9% in patients admitted for COPD exacerbation [211].

Other oral hyperglycemic drugs, such as thiazolidines (TZDs), act on master transcriptional inflammation regulators, namely PPARy, and function as anti-inflammatory and anti-atherogenic agents [212]. PPARy agonists have shown promising results in the treatment of airway neutrophilia [213] and COPD [214], in addition to their management of diabetes. TZDs are associated with a reduced risk of COPD exacerbations [199]. The use of another anti-inflammatory diabetes therapy, PDE4 inhibitors, was also tested for the treatment of COPD exacerbation frequency [215].

### 6.3. Dietary Links and Lung Function

The pattern of dietary intake is an important factor in the pathogenesis and prevention of COPD and diabetes, especially since obesity is a risk factor and a comorbidity of both. Dietary factors could indirectly alter the impact of adverse environmental exposures or genetic predisposition in COPD and diabetes, and may also directly affect biological processes associated with lung function, disease development, inflammation, and outcomes. Studies using self-reported data on dietary intake are informally known to be significantly inaccurate, and the collected data may be associated with additional lifestyle choices that can impact the outcome of the study. Therefore, it is difficult to propose mechanistic links for the available data. Lower energy intake, with varying intake of macro and micro nutrients, are observed in COPD patients [216] in combination with obesity [217]. Several recent large population-based prospective studies showed an inverse and independent association between the long-term consumption of fruits and vegetables and the incidence of COPD [218,219]. Fiber intake is also observed as a risk factor for COPD [220]. Equally, dietary intakes of vitamin C, a hydrophilic antioxidant, was reported to be associated with FEV1 preservation [221]. Similarly, vitamin D supplementation trials to prevent COPD exacerbation suggest that it may only benefit patients with low baseline vitamin D levels [222]. In a recent randomized trial in COPD patients, supplementation with flavonoids (oligomeric pro-anthocyanidins extracted from grape seeds) improved oxidative stress and lipid profile, but not lung function [223]. A case-control study in Japanese adults observed a positive association between the intake of calcium, phosphorus, iron, potassium, and selenium with FEV1 measurements [224]. Polyunsaturated fatty acids of the omega-3 family, such as α-linolenic acid, eicosapentaenoic acid, and docosahexaenoic acid, may play an anti-inflammatory role in COPD and diabetes, but randomized controlled trials are needed to confirm any relationships between the intake of polyunsaturated fatty acids and COPD, with or without diabetes. A comprehensive review of the role of dietary metabolites and dietary treatment in pulmonary function in COPD is outlined in the cited review paper [225]. Several recent studies also suggest a role of extracellular cholesterol and impaired cholesterol efflux in pulmonary outcomes [226]. This topic will be an exciting area for further research in COPD, diabetes, and other comorbidities.

## 7. Conclusions

There appears to be mounting evidence of common signaling and genetic signature links between COPD, T1D, and T2D. Promisingly, treatments for these isolated conditions seem to have broad-acting effects that ameliorate COPD and diabetes symptoms and slow disease progression. Given the alarmingly increasing burden of COPD and diabetes worldwide, identification of modifiable risk factors, intervention options, and novel therapeutic options are of interest.

## Figures and Tables

**Figure 1 medicina-58-01030-f001:**
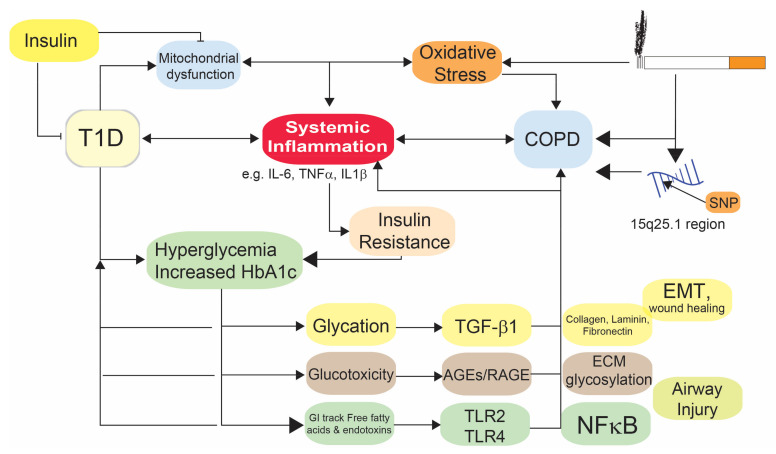
Possible mechanisms resulting in the increased prevalence of T1D in COPD.

**Figure 2 medicina-58-01030-f002:**
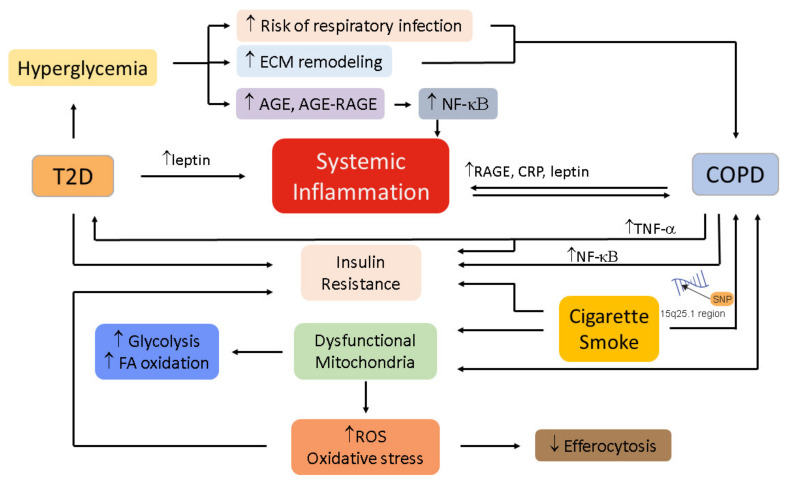
Possible mechanisms linking T2D to COPD.

**Table 1 medicina-58-01030-t001:** Summary of therapeutic strategies for COPD and diabetes.

Therapy	Mechanism of Action	Outcomes in Patients with T2D	Outcomes in Patients with COPD	Outcomes in Patients with COPD and T2D
Metformin	Inhibits proinflammatory NF-kB signaling in human vascular cells [178].	Reduces the formation of AGEs by improving glycemic control [179]. Reduces CV mortality, all-cause mortality, and CV events in T2D patients with CAD [180]. Changes the composition of gut microbiota, improves insulin resistance, and decrease tissue inflammation [181].	Reduces inflammation in the elastase-induced emphysema mouse model [179]. The reduction in lung infections modifies glucose flow across the lung tissue [182,183]. Improves survival rates in COPD patients with T2D [184]. Reduces loss of CO diffusing capacity [185].	Improves health, symptoms, hospitalizations, and mortality in patients with COPD and T2D [186]. Reduces the likelihood of T2D developing COPD [187] T2D with moderate to severe COPD with symptomatic improvement measured in SGRQ and TDI [188]. COPD patients with T2D are at higher risk of pneumonia, COPD exacerbation, and need for mechanical ventilation [189].
Inhaled corticosteroids	Lower the migration of inflammatory cells; reverse capillary permeability and lysosomal stabilization [190].	Dose-dependent elevation in serum glucose concentration [191] and increase in HbA1C [192].	Increased risk of new-onset T2D [193,194]. 34% increase in the rate of diabetes and increased rate of diabetes progression [195]. No increased risk of diabetes among current users [196].	
Thiazolidinediones	Increase insulin sensitivity by binding and activating PPARs, altering the transcription of glucose and lipid metabolism-related genes [197].	Increase glucose utilization in the tissues by increasing insulin sensitivity [198].	Associated with reduced risk of COPD exacerbations [199]	
AAT augmentation	Physiologic AAT inactivates proteolytic enzymes secreted during inflammation and has anti-apoptotic properties [200].	Safe and well-tolerated in stage 3 T1D [45]. Reduced HbA1c levels in adolescents with recently diagnosed T1D [47]. Children with AAT infusions showed fewer IL-1β producing monocytes and dendritic cells [45].	AAT augmentation slows the progression of emphysema [201].	

AGEs: Advanced glycation end products; SGRQ: St. George’s Respiratory Questionnaire; TDI: transitional dyspnea index.

## Data Availability

Not applicable.

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
