# Peer review of "Mechanisms Linking COPD to Type 1 and 2 Diabetes Mellitus: Is There a Relationship between Diabetes and COPD?"

_medicina, 2022, doi:10.3390/medicina58081030_

Round 1
Reviewer 1 Report
This review article summarized the mechanisms linking COPD to T1D and T2D. Overall, this is an interesting and timely review of the field. I have several comments for the authors to address.
1. Sections 2.2, 2.3 and 5.2
As the authors noted, metabolic syndrome and chronic inflammation are the important mechanisms underlying the association between COPD and T2D. An important common risk factor for both diseases is obesity. Obesity is considered a chronic low-grade inflammation and also central obesity has a direct mechanical effect on lung function. On lines 96-99, the authors mentioned central obesity after adjusting BMI was associated with airflow obstruction. What are the underlying mechanisms for this association? For example, shared genetics may explain the mechanism. Please read the following 2 papers and add a discussion of mechanisms in Section 5.2 based on the following papers.
Zhu Z, Guo Y, Shi H, Liu CL, Panganiban RA, Chung W, O'Connor LJ, Himes BE, Gazal S, Hasegawa K, Camargo CA Jr, Qi L, Moffatt MF, Hu FB, Lu Q, Cookson WOC, Liang L. Shared genetic and experimental links between obesity-related traits and asthma subtypes in UK Biobank. Journal of Allergy and Clinical Immunology. 2020;145(2):537-549.
Zhu Z, Li J, Si J, Ma B, Shi H, Lv J, et al. A large-scale genome-wide association analysis of lung function in the Chinese population identifies novel loci and highlights shared genetic aetiology with obesity. European Respiratory Journal. 2021 14;58(4):2100199.
2. Section 2.4 and 5.3
The authors mentioned cigarette smoking is a potential common risk factor for COPD and T2D. This discussion is interesting. Have the authors thought about the genetic contribution to this? For example, an important smoking-related genetic locus is 15q25.1. Can this locus contribute to both COPD and T2D? Please see the following paper as an example.
Zhu Z, Wang X, Li X, Lin Y, Shen S, Liu CL, Hobbs BD, Hasegawa K, Liang L; International COPD Genetics Consortium, Boezen HM, Camargo CA Jr, Cho MH, Christiani DC. Genetic overlap of chronic obstructive pulmonary disease and cardiovascular disease-related traits: a large-scale genome-wide cross-trait analysis. Respir Res. 2019;20(1):64.
3. Finally, in terms of the environmental contribution, I recommend the authors add a discussion about the effect of diets on COPD and T2D. For example, what kind of dietary metabolites can lead to or protect both diseases, Omega-3 PUFA, a low-carb diet?
Author Response
Reviewer 1:
- Sections 2.2, 2.3 and 5.2
As the authors noted, metabolic syndrome and chronic inflammation are the important mechanisms underlying the association between COPD and T2D. An important common risk factor for both diseases is obesity. Obesity is considered a chronic low-grade inflammation and also central obesity has a direct mechanical effect on lung function. On lines 96-99, the authors mentioned central obesity after adjusting BMI was associated with airflow obstruction. What are the underlying mechanisms for this association? For example, shared genetics may explain the mechanism. Please read the following 2 papers and add a discussion of mechanisms in Section 5.2 based on the following papers.
Zhu Z, Guo Y, Shi H, Liu CL, Panganiban RA, Chung W, O'Connor LJ, Himes BE, Gazal S, Hasegawa K, Camargo CA Jr, Qi L, Moffatt MF, Hu FB, Lu Q, Cookson WOC, Liang L. Shared genetic and experimental links between obesity-related traits and asthma subtypes in UK Biobank. Journal of Allergy and Clinical Immunology. 2020;145(2):537-549.
Zhu Z, Li J, Si J, Ma B, Shi H, Lv J, et al. A large-scale genome-wide association analysis of lung function in the Chinese population identifies novel loci and highlights shared genetic aetiology with obesity. European Respiratory Journal. 2021 14;58(4):2100199.
Response: We thank the reviewer for their constructive feedback and have tried to address all your concerns/comments. The central obesity and BMI association are not fully discussed in the paper outlined here (Lam et al., ERJ 2010). However, we have discussed both recommended papers and incorporated this genetic approach into several sections. Please see lines 352-365.
- Section 2.4 and 5.3
The authors mentioned cigarette smoking is a potential common risk factor for COPD and T2D. This discussion is interesting. Have the authors thought about the genetic contribution to this? For example, an important smoking-related genetic locus is 15q25.1. Can this locus contribute to both COPD and T2D? Please see the following paper as an example.
Zhu Z, Wang X, Li X, Lin Y, Shen S, Liu CL, Hobbs BD, Hasegawa K, Liang L; International COPD Genetics Consortium, Boezen HM, Camargo CA Jr, Cho MH, Christiani DC. Genetic overlap of chronic obstructive pulmonary disease and cardiovascular disease-related traits: a large-scale genome-wide cross-trait analysis. Respir Res. 2019;20(1):64.
Response: We thank the reviewer for their excellent suggestion and we have discussed this topic on lines 142-155.
- Finally, in terms of the environmental contribution, I recommend the authors add a discussion about the effect of diets on COPD and T2D. For example, what kind of dietary metabolites can lead to or protect both diseases, Omega-3 PUFA, a low-carb diet?
Response: We have added another section for this suggested topic. This is a large topic for discussion but we have tried to limit the depth of this section and referenced other complimentary reviews and papers to expand on the scope of the topic. Please see section 6.3 on lines 575-604.
Reviewer 2 Report
The title is also confusing: metabolic syndrome is not the same thing as DM. The authors should focus on one or the other.
The abstract is too general. Conclusions should be mentioned. In fact this is what misses in the whole article: the critical appraisal of the literature. Without it, the value of the article consists only in an extensive inventory – which is good - but not enough. It looks like an inventory not a revision of the literature and mainly creates confusion. I fully understand that there are many fields in which existing data are not conclusive but the added value of a review is the authors’ contribution in analyzing the literature (gaps, possible explanations of contradictory findings etc). Because otherwise, with only general comments extracted from the references, one has to read the reference in order to understand the findings in their real context, particularly if these findings are presented without their limitations.Therefore, the manuscript needs major revision and improvement before publication.
Beside this general observation, I will briefly mention some aspects which should be improved:
As the mechanisms linking T1DM and T2DM to COPD are different, it is preferable to present the epidemiological and the experimental data separately. The way they are mixed up now creates confusion.
Line 55-56. The hyperglycemia in acute patients is not necessarily a marker of diabetes. The baseline glycemia as predictor of clinical outcomes is not an argument for this review. Should be reformulated.
Line 105. The reference 31 is not a consensus, but an original article. Other source should be cited.
Line 104. The reference 26; the groups differ too much in terms of number of pack years, a factor which is a more plausible explanation about the difference in lung obstruction than the presence of the metabolic syndrome. No adjustments were made and therefore the conclusion is somehow speculative.
Lines 103-107: it is confusing: Metabolic syndrome aggravates or is associated with better lung function (earlier stages or aggravation)? If the results are different, some explanation should be presented.
Lines 253-256 would better fit in the description of the decline function (clinical consequences of T1DM chapter :epidemiological evidence
Line 258; the oxidative stress is present also in T1DM. Why is is mentioned only as mechanism in T2DM?
Lines 314-315 and 320-321 – they repeat almost the same idea. If there is a “suggestion”, the mechanism underlying this relation should be mentioned.
Line 322-326 eg. the association (pozitive or negative) between adiponectin and lung function. What made the difference between the studies with conflicting results: the sample, the type of adiponectin measured, the age, other interrelations?
338-340 a chapter is dedicated to the oxidative stress and this mechanism could be included in that part.
386-387: Reference 160 does not describe the mechanism, but refers to another study, which should be presented here.
Fig 2: adiponectin, as previously mentioned is not always increased in the metabolic syndrome, the low adiponectin and high leptin are generally associated with the inflammation.
The consequence of high AGE in DM and increased expression of RAGE in COPD was explored or is just a another possible interaction?
As mentioned in the 159 reference RAGE is abundant also in normal lung. The consequence of high AGE in DM and increased expression of RAGE in COPD was explored or is just another possible interaction?
Author Response
Reviewer 2:
The title is also confusing: metabolic syndrome is not the same thing as DM. The authors should focus on one or the other.
Response: We agree with the review and modified the title
The abstract is too general. Conclusions should be mentioned. In fact this is what misses in the whole article: the critical appraisal of the literature. Without it, the value of the article consists only in an extensive inventory – which is good - but not enough. It looks like an inventory not a revision of the literature and mainly creates confusion. I fully understand that there are many fields in which existing data are not conclusive but the added value of a review is the authors’ contribution in analyzing the literature (gaps, possible explanations of contradictory findings etc). Because otherwise, with only general comments extracted from the references, one has to read the reference in order to understand the findings in their real context, particularly if these findings are presented without their limitations. Therefore, the manuscript needs major revision and improvement before publication.
Response: We appreciate and agree with the recommended meta-analysis approach suggested by the reviewer. However, we believe this format is not suited to this current manuscript at this moment in time, as the majority of mechanistic studies on this topic are in animals and there are insufficient clinical trials performed to conduct a meta-analysis systemic review on this topic. We have tried to address the majority of your excellent comments in the 7 days allotted for changes. We really appreciate the input of the reviewer and excellent comments.
Beside this general observation, I will briefly mention some aspects which should be improved:
As the mechanisms linking T1DM and T2DM to COPD are different, it is preferable to present the epidemiological and the experimental data separately. The way they are mixed up now creates confusion.
Response: We have made these suggested changes. Please see section 2.1 on lines 50-58.
Line 55-56. The hyperglycemia in acute patients is not necessarily a marker of diabetes. The baseline glycemia as predictor of clinical outcomes is not an argument for this review. Should be reformulated.
Response: The primary diagnosis of patients outlined in the referenced paper on these lines was COPD patients. We do not state that this was a marker for diabetes in this sentence. We have tweaked this sentence. Please see line 73.
Line 105. The reference 31 is not a consensus, but an original article. Other source should be cited.
Response: We have modified the wording of this sentence. Please see lines 128-130.
Line 104. The reference 26; the groups differ too much in terms of number of pack years, a factor which is a more plausible explanation about the difference in lung obstruction than the presence of the metabolic syndrome. No adjustments were made and therefore the conclusion is somehow speculative.
Response: We have removed this reference as suggested. Please see lines 127-130.
Lines 103-107: it is confusing: Metabolic syndrome aggravates or is associated with better lung function (earlier stages or aggravation)? If the results are different, some explanation should be presented.
Response: We agree with the reviewer that these sentences were a little confusing and we have adjusted these sentences. Please see lines 127-130.
Lines 253-256 would better fit in the description of the decline function (clinical consequences of T1DM chapter :epidemiological evidence
Response: We agree with the reviewer and have moved these sentences. Please see section 2.1 on lines 50-58.
Line 258; the oxidative stress is present also in T1DM. Why is is mentioned only as mechanism in T2DM?
Response: We agree with the reviewer and added a separate oxidative stress section to the T1D mechanism section. Please see section 4.1 on lines 200-216.
Lines 314-315 and 320-321 – they repeat almost the same idea. If there is a “suggestion”, the mechanism underlying this relation should be mentioned.
Response: We have modified these sentences. Please see lines 302-303.
Line 322-326 eg. the association (pozitive or negative) between adiponectin and lung function. What made the difference between the studies with conflicting results: the sample, the type of adiponectin measured, the age, other interrelations?
Response: We agree with the reviewer and have added additional information on the limitations of these studies. Please see lines 349-351 and 402-405.
338-340 a chapter is dedicated to the oxidative stress and this mechanism could be included in that part.
Response: We have now added another section on oxidative stress. Please see lines 234-253
386-387: Reference 160 does not describe the mechanism, but refers to another study, which should be presented here.
Response: We have replaced this reference. Please see section 4.1 on lines 200-216.
Fig 2: adiponectin, as previously mentioned is not always increased in the metabolic syndrome, the low adiponectin and high leptin are generally associated with the inflammation.
Response: We have removed the adiponectin from figure 2.
The consequence of high AGE in DM and increased expression of RAGE in COPD was explored or is just a another possible interaction?
Response: This is just a possible interaction. Please see lines 461-465.
As mentioned in the 159 reference RAGE is abundant also in normal lung. The consequence of high AGE in DM and increased expression of RAGE in COPD was explored or is just another possible interaction?
Response: This is just a possible interaction. Please see lines 461-465.
Round 2
Reviewer 1 Report
The authors have addressed well to my comments.
Reviewer 2 Report
The authors have replied to my comments.